# Brief Communication: Subglacial lake drainage beneath Isunguata Sermia, West Greenland: geomorphic and ice-dynamic effects

**Stephen J. Livingstone[1], Andrew J. Sole[1], Robert D. Storrar[2], Devin Harrison[3], Neil Ross[3], Jade Bowling[4]**

[1]Department of Geography, University of Sheffield, Sheffield, UK

[2]Department of the Natural and Built Environment, Sheffield Hallam University, UK

[3]School of Geography, Politics and Sociology, Newcastle University, Newcastle upon Tyne, UK

[4]Lancaster Environment Centre, Lancaster University, Lancaster, UK

*Correspondence to*: Stephen J. Livingstone (s.j.livingstone@sheffield.ac.uk)

**Abstract (100 words)**

We report three active subglacial lakes within 2-km of the lateral margin of Isunguata Sermia, West Greenland, identified by differencing time-stamped ArcticDEM strips. Each lake underwent one drainage-refill event between 2009 and 2017, with two lakes draining in <1 month in August 2014 and August 2015. The 2015 drainage caused a ~1-month down-glacier slowdown in ice-flow and flooded the foreland, aggrading the proglacial channel by 8 m. The proglacial flooding confirms the ice-surface elevation anomalies as subglacial water bodies and demonstrates how their drainage can significantly modify proglacial environments. These subglacial lakes offer accessible targets for geophysical investigations and exploration.

## 1. Introduction

Meltwater beneath the Greenland Ice Sheet is sourced from geothermal and frictional melting, and via the input of surface meltwater through englacial pathways. This meltwater drains towards the ice sheet margin through a complex network of inefficient and efficient drainage routes. Spatial and temporal variations in drainage structure are controlled by the hydraulic gradient and meltwater flux. Steeper hydraulic gradients and higher meltwater fluxes close to the ice margin lead to greater channel melt rates and promote the formation of efficient channels, which can extend up to 40 km inland and evolve on seasonal timescales in response to surface meltwater inputs (Chandler et al., 2013). Low hydraulic gradients and smaller meltwater fluxes dominated by subglacial meltwater sources tend to be associated with more inefficient drainage configurations further inland (Doyle et al., 2014).

Storage of water in firn (Forster et al., 2013), damaged englacial ice (Kendrick et al., 2018) and both supraglacial (Selmes et al., 2011) and subglacial lakes (Palmer et al., 2013; Bowling et al., 2019) can delay the drainage of meltwater through the ice sheet to the ocean, while the rapid drainage of stored water can overwhelm the drainage system and perturb ice flow (e.g. Das et al., 2008). Storage and drainage of supraglacial lakes have been well-documented (e.g. Selmes et al., 2011), but the volume of water stored subglacially, the residence times of the lakes, and the wider influence on the subglacial hydrological system and ice flow is poorly understood. Although subglacial lakes are expected to be a less important component of the hydrological system compared with Antarctica (e.g. Siegfried & Fricker, 2018) due to steeper hydraulic gradients, dominance of surface meltwater inputs and more efficient subglacial water routing, 1000s of subglacial lakes have been predicted and over 50 identified beneath the Greenland Ice Sheet (Livingstone et al., 2013; Bowling et al., 2019). This includes stable lakes above the Equilibrium Line (EL) but away from the interior identified from airborne radio-echo sounding (Palmer et al., 2013; Bowling et al., 2019); hydrologically active lakes recharged by surface meltwater near the EL, determined from surface elevation change measurements derived from repeat high-resolution Digital Surface Models (DSMs) and ICESat elevation data (Palmer et al., 2015; Willis et al., 2015; Bowling et al., 2019); and

small seasonally active subglacial water bodies below the EL which form during winter and drain during the melt season identified from repeat airborne radio-echo sounding (Chu et al., 2016).

Whilst seasonal water storage is thought to be common below the EL (e.g. Chu et al., 2016; Kendrick et al., 2018), longer term subglacial lake storage is thought unlikely due to the development of efficient channels and the associated increase in hydrological connectivity each melt season (Bowling et al., 2019). In this paper we acquired multi-temporal ArcticDEM DSMs (Noh & Howat, 2015) and Landsat 7 and 8 satellite imagery between 2009 and 2017 (distributed by the U.S. Geological Survey - https://earthexplorer.usgs.gov/) to identify three active subglacial lakes on a reverse bed-slope beneath Isunguata Sermia, West Greenland (67°10' N, 50°12'W) (Fig. 1).

## 2. Observations

Yearly ice-surface elevation change was determined from 2009 to 2017 by differencing the annually-averaged ArcticDEM DSMs. This revealed three distinctive quasi-circular features, hereafter referred to as 'anomalies', all within 2 km of the lateral margin of Isunguata Sermia, that were characterised by periods of subsidence followed by uplift (Fig. 1a). Timeseries of relative elevation change for each anomaly, which are not advected towards the margin, were calculated from sub-annual ArcticDEM DSMs by subtracting the mean ice-surface elevation of the anomaly from the mean elevation of a 500 m buffer around it (Fig. 1b). This approach was used to isolate the dynamic effect and to remove the influence of systematic vertical and horizontal offsets (of up to 3-5 m) between DSMs. Anomaly 1, located <5 km from the terminus of Isunguata Sermia where the ice is ~325 m thick (Lindbäck et al., 2014), formed a 0.93 km$^2$ depression between 19$^{th}$ August 2010 and 3$^{rd}$ August 2011 with a mean depth of 5 m and maximum depth of 17 m. The ice-surface then rose 1 m by November 2011 before recovering back to its 2010 elevation by February 2013. Anomaly 2 (~370 m ice thickness), about 1 km further up ice, formed a 0.88 km$^2$ depression between 2$^{nd}$ August 2015 and 21$^{st}$ September 2015, with a mean depth of 13 m and maximum depth of 30 m. It then rose 9 m between 2015 and 2017. Anomaly 3 (~450 m ice thickness), immediately up-ice of anomaly 2 and ~9 km from the terminus, formed a 0.67 km$^2$ depression between 17$^{th}$ August 2014 and 19$^{th}$ September 2014, with a mean depth of 4 m and maximum depth of 14 m, before the surface rose 3 m between 2014 and 2017. Surface structural features indicative of localised ice fracture during subsidence, such as crescentic crevasses, are not apparent in any of the depressions.

Ice velocities derived from feature tracking of Sentinel-1 radar data (SI methods) do not show a consistent area of fast flow over any of the anomalies. However, during the subsidence of anomaly 2 in late July to early August 2015 (Fig. 2), ice flow immediately downstream of the anomaly experienced a net slowdown to roughly winter values (Fig. 2b-c), coincident with an overall regional slowdown in ice flow. This was followed by a return to pre-subsidence flow speeds by 19$^{th}$ – 31$^{st}$ August 2015 (Fig. 2d). During the period of subsidence ice flow also converged into the depression, leading to localised rapid 'up-glacier' flow against the regional westerly flow direction, which manifests as a strong positive ice flow change immediately above the anomaly (Fig. 2c).

Landsat 8 OLI satellite images acquired before and after the 2015 ice-surface subsidence (anomaly 2) reveal a major change in the 1.8 km wide proglacial braided river system (Fig. 3). On 15$^{th}$ July 2015 the river plain is characterised by a single channel emanating from the front of Isunguata Sermia, that then bifurcates down-river into multiple braids and intervening bars (Fig. 3a). Dry areas above the water level are demarcated by a sharp change in colour, with wetted areas darker and dry areas lighter. Using this demarcation, a major flood plain directly in-front of the main portal, which causes the river emanating from the glacier to divert northwards and then westwards, is identified. On the basis of a qualitative change from light to dark, on the 25$^{th}$ August 2015, and a quantified positive change in Normalised Difference Water Index (NDWI) (SI methods) of up to +0.23 (mean: +0.09) between July and August, the dry areas (bars and floodplain) became inundated by water and the braided river system re-organised (Fig. 3b-c). Differencing ArcticDEM DSMs of the proglacial area before (4$^{th}$ May 2015) and after (21$^{st}$ September 2015) the ice-surface elevation change of August-September 2015 associated with anomaly 2 reveals 3 m of mean net sediment aggradation across a 5 km stretch of the main proglacial channel (Fig. 3d). Aggradation was up to 8 m close to the outlet and declined to <1 m 5 km from the glacier terminus.

## 3. Discussion

We identify three ice surface elevation anomalies on Isunguata Sermia, which we interpret as subglacial lake drainage and filling (Fig. 1). From this point onwards we therefore refer to these anomalies as 'Subglacial Lakes 1-3'. Confirmation of a subglacial lake origin is provided by flooding of the proglacial outwash plain in August 2015, which coincided with the timing of ice-surface elevation anomaly 2, evidencing the release of meltwater (Fig. 3). This inundation (wetting) of the flood plain is not replicated at the nearby Leverett-Russell Glacier (Fig. 3b), ruling out a common external forcing (e.g. heavy rainfall). All three subglacial lakes are located under 325-450 m thick (Lindbäck et al., 2014), warm-based ice on a reverse gradient bed slope (15 m per km); the reverse slope may be trapping the water causing the lakes to form. Although subglacial hydrological analysis (e.g. Chu et al., 2016) does not produce hydraulic minima in the locations where we identify lakes, this is likely to be the result of the limited and relatively poor-quality airborne radar ice-thickness measurements across the thin, near-marginal area of Isunguata Sermia, and the resolution of the bed DEMs.

The three subglacial lakes underwent one drainage event each over the 8-year data period (Fig. 2). Differencing of the DSMs either side of the drainage events, over the area of the ice-surface anomalies, gives total lake volume changes of $6.5 \pm 0.52 \times 10^6$ m$^3$, $1.3 \pm 0.05 \times 10^7$ m$^3$ and $3.5 \pm 0.38 \times 10^6$ m$^3$ for anomalies 1-3 respectively. Drainage of Subglacial Lake 2 in 2015 and Lake 3 in 2014 were both triggered in August and drained in <1 month, which is consistent with other larger subglacial lake drainage events identified in Greenland (Palmer et al., 2015; Willis et al., 2015; Bowling et al., 2019), but contrasts with the longer (months to years) drainage period of those in Antarctica (e.g. Siegfried & Fricker, 2018). If the vertical displacement of the ice-surface is equivalent to the depth of the subglacial lake, this gives a mean minimum discharge of 6.5 m$^3$ s$^{-1}$ for Subglacial Lake 2, which is the largest and best-constrained lake, by available DSM and satellite imagery, in this study. This is, however, well below the likely maximum discharge given the enormous mobilisation of sediment that resulted from the drainage and the unknown period of drainage within the observational window.

Lake recharge is on the scale of a few years, and it is noticeable that the largest subglacial lake drainage event (Lake 2) subsequently refilled at the fastest rate (~5 m yr$^{-1}$ uplift), while the smallest drainage event (Lake 3) filled at the slowest rate (~1 m yr$^{-1}$ uplift). The lakes are at the lower end of the ablation zone and therefore would be expected to be dominated by upstream surface meltwater inputs and the seasonal melt signal. Despite this, lake drainage is not associated with high-melt years (e.g. the 2011 drainage event coincided with a low melt year). All three lakes exhibited quiescent periods of extended high-stand, which might occur when water flow into the lake is balanced by outflow, and suggests an external threshold controlling lake drainage initiation.

Although the three subglacial lakes are in close proximity, drainage of an upstream lake does not trigger a cascade of drainage in downstream lakes. In addition, the filling of Lake 3 did not limit recharge of Lake 2 just downstream (Fig. 1b). This suggests that the lakes are not hydraulically well connected, consistent with subglacial hydraulic routing analysis, which indicates that the main subglacial drainage axis is just to the north of the two upstream subglacial lakes (Fig. 3a). Both the 2014 and 2015 drainage events were initiated in August at a time when drainage system connectivity is envisaged to be high and water preferentially drains towards efficient channels along a hydraulic gradient. Thus, rapid drainage could be a response to lakes infrequently connecting with the main subglacial channel.

Drainage of Subglacial Lake 2 in August 2015 resulted in a net slowdown in downstream ice flow of Isunguata Sermia over a short period of about one month, before ice flow subsequently returned to pre-drainage ice-flow speeds in late August (Fig. 2). This slowdown occurs abruptly at the location of lake 2 suggesting its drainage has impacted ice flow (Fig. 2b-c), but is also coincident with a broader regional slowdown that must at least partly overprint the dynamic signal of the subglacial lake. Although we cannot rule out a short-term speed-up in response to initial drainage, any acceleration must have been extremely rapid (< the twelve-day repeat period imagery used to calculate the velocities), and of smaller overall magnitude and/or duration than the subsequent slowdown in ice flow. This response to subglacial lake drainage is the opposite of temporary ice accelerations observed in Antarctica (e.g. Scambos et al. 2011), and is likely because any initial speed-up would have been dampened by efficient drainage of the lake through existing large subglacial channel(s) near the ice margin. Melting of the channel sides and erosion of the channel bed caused by turbulent water flow would have enlarged the channel's cross-sectional area leading to reduced water pressure, likely causing deceleration of overlying ice. This is based on the timing of the event at the end of the melt season, the position of the lake near to the ice-margin (Fig. 1) and the large subglacial drainage catchment of Isunguata Sermia (Chu et al., 2016), all of which would act to promote the development of large and efficient channels (e.g. Chandler et al., 2013). This inference is supported by the pattern of thickest sediment deposition at the southern end of the glacier terminus (Fig. 3), which suggests

that the subglacial drainage event was at least partially focused into a channel rather than an unconstrained sheet flood.

The August 2015 subglacial lake drainage event flooded the foreland and resulted in substantial net ice-proximal sediment aggradation ($7.5 \times 10^6$ m$^3$) of the outwash plain (Fig. 3). Deposition was greatest in the main channel, with up to 8 m of net aggradation close to the outlet reducing to <1 m 5 km from the terminus. This near-margin pattern of aggradation is consistent with the geomorphic impact of jökulhlaups observed in Iceland (e.g. Dunning et al., 2013) and demonstrates the potential of episodic subglacial lake drainage events to erode the subglacial bed and modify the proglacial environment. Given the subglacial lake is located just 8 km from the glacier terminus the subglacial erosion necessary to produce the sediment volume deposited on the foreland is equivalent to a 10 m deep and 100 m wide channel cut into the bed.

The three subglacial lakes beneath Isunguata Sermia do not exhibit the seasonal pattern of winter storage and summer drainage that was previously thought to dominate in the ablation zone (Chu et al., 2016; Bowling et al., 2019). Instead, they share many of the drainage characteristics of the hydrologically active subglacial lakes identified near the EL of the Greenland Ice Sheet (e.g. Palmer et al., 2015; Willis et al., 2015; Bowling et al., 2019). This indicates the potential for multi-year storage of subglacial water towards the margin of the ice sheet in regions dominated by surface meltwater inputs to the bed. This storage could delay the delivery of meltwater to the margin and influence the ice dynamic response to surface meltwater downstream.

Although the presence or absence of sediments in these lakes or the thickness of any water column has yet to be tested, these three subglacial lakes present an extremely accessible target for future geophysical characterisation and active lake exploration. Ice-surface elevation changes suggest the lakes are at least 14-30 m deep and have minimum volumes of $3.5$-$13 \times 10^6$ m$^3$. Ice cover is relatively thin (325-450 m), and the lakes are clustered and in close proximity to the ice margin (<2 km), road (<5 km) and key logistical support including a major airport (Kangerlussuaq). Key questions that could be addressed through detailed investigation of these lakes include: what triggers subglacial lake drainage and how does drainage evolve downstream? How do the lakes interact with other components of the subglacial drainage system? What geomorphological and sedimentological signatures of similar drainage events might be recorded in the proglacial area?

## 4. Conclusions

Using multi-temporal ArcticDEM DSMs and satellite imagery, we identify three active subglacial lakes <10 km from the terminus of Isunguata Sermia. The lakes are characterised by periods of relative inactivity punctuated by rapid drainage (<1 month) and then slow recharge (a few years). The most recent drainage event in 2015 flooded the outwash plain resulting in net ice-flow slowdown over a ~1 month period and net ice-marginal sediment aggradation that was greatest closest to the outflow portal and thinned downstream. This work demonstrates the potential for subglacial lakes to exist in the lower ablation zone close to the ice margin, where subglacial hydrology is dominated by seasonal surface meltwater inputs and efficient channelized drainage. The lakes appear to be only weakly connected to the main subglacial channel axis and drainage may be controlled by the ability of this channel to occasionally capture water from its surroundings. The 2015 subglacial lake drainage event had a subglacial and proglacial geomorphic impact, including substantial erosion of sediment from beneath Isunguata Sermia and extensive aggradation of sediment in the proglacial outwash plain close to the terminus. Net slowdown in ice flow due to subglacial lake drainage is likely due to antecedent subglacial channelized drainage close to the margin towards the end of the melt season that at least partially accommodated the lake outburst flood and therefore dampened any initial acceleration. Melting of the channel sides and erosion of its bed caused by the turbulent water flow would have enlarged the channel's cross-sectional area leading to reduced water pressure which likely caused the deceleration of overlying ice. This suggests that the ice-dynamic response to subglacial lake drainage may vary depending on their subglacial and englacial context. Detailed geophysical studies across and downstream of these lakes would provide insight into the conditions causing the lakes to form and drain, the resultant geomorphic imprint and the depositional archive of these lake environments. Crucially, these subglacial lakes may be the most accessible in the world due to their setting beneath thin ice close to the lateral margin of the glacier and the existing infrastructure and logistical set-up of the region.

*Author contributions.* A. Sole and R. Storrar initially identified the subglacial lakes. S. Livingstone, A. Sole and D. Harrison processed and analysed the data. S. Livingstone wrote the paper with input and ideas from all co-authors.

### *Appendix A. Datasets and Methods.*

The ArcticDEM DSMs were generated from high-resolution satellite imagery and have a horizontal spatial resolution of 2 m and internal accuracy of 0.2 m. Each of the 52 DSMs acquired over the time period were corrected against filtered IceSAT altimetry data using the metadata provided (Dai & Howat, 2017). Minimum subglacial lake volumes were calculated by differencing DSMs either side of the drainage events, over the area of the ice-surface anomalies, with uncertainties derived by multiplying the internal error of the ArcticDEM by the
lake area both before and after drainage and adding together in quadrature.

Normalised Difference Water Index (NDWI) analysis is an effective method for highlighting water bodies and saturated environments whilst subduing background information. Before NDWI calculations were undertaken Landsat 8 OLI images were pre-processed to convert raw pixel digital number (DN) values into top of atmosphere (TOA) reflectance (Zhao et al., 2018). Change in NDWI to identify flooding of the proglacial zone was calculated
using the TOA corrected Landsat 8 green (band 3) and near-infrared (band 5) bands and the formula: NDWI = (band 3 – band 5)/(band 3 + band 5) (Zhao et al., 2018).

Ice velocity was calculated from feature and speckle tracking of Sentinel 1a Interferometric Wide Swath mode Single-Look Complex Synthetic Aperture Radar amplitude images following the approach outlined in Tuckett et al. (2019). This included cross-correlation between repeat-pass image pairs to determine the offset of features
(e.g. crevasses) over time and processing involved co-location of image pairs using precise satellite orbit ephemerids, conversion of images to amplitude in GMTSAR, a Butterworth high-pass filter to remove image brightness variations with a wavelength greater ~1 km and tracking of images in MATLAB using PIVSuite (https://uk.mathworks.com/matlabcentral/fileexchange/45028-pivsuite) adapted for quantifying ice flow. These data cover one swath and include 12 twelve-day repeat image pears between 3$^{rd}$ January and 31$^{st}$ August 2015.
Anomalies were calculated relative to the period 3rd January – 5th April.

*Acknowledgements.* DEMs provided by the Polar Geospatial Center under NSF-OPP awards 1043681, 1559691, and 1542736. This work was supported by Ph.D. studentships awarded to D. Harrison through the IAPETUS Natural Environmental Research Council Doctoral Training Partnership (NE/L002590/1) and J. Bowling through the ENVISION Natural Environmental Research Council Doctoral Training Partnership
(EAA6583/3152).*

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

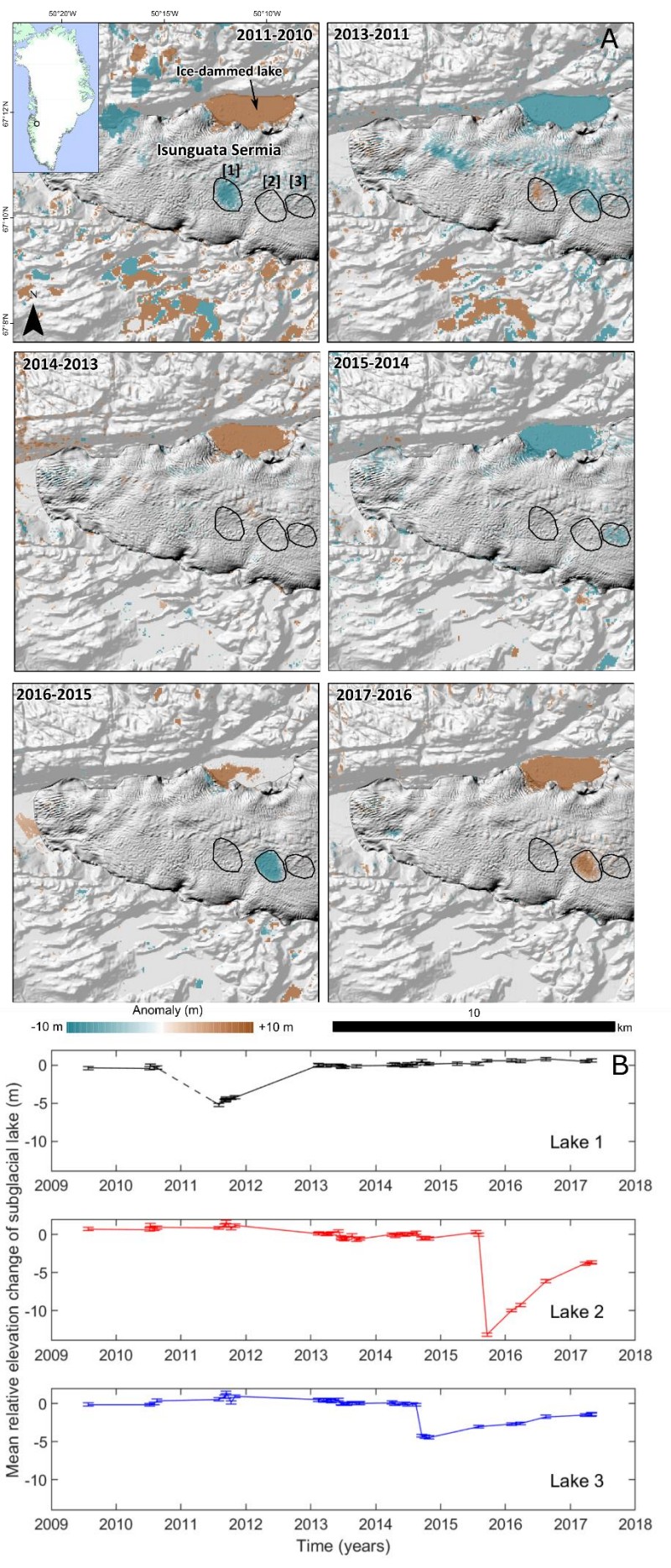

**Figure 1:** Ice surface elevation change. A. Yearly anomaly plots of ice-surface elevation change from 2010-2017 annually averaged DSMs, calculated as the median of all available timestamped ArcticDEM tiles in that particular year down-sampled to 50 m. Black outlines and numbers (1-3) reference the location of three identified subglacial lakes. Elevation anomalies have also picked out the filling and drainage of an ice-dammed lake at the northern margin of Isunguata Sermia. Anomalies <5 m have been removed. The background image is an ArcticDEM DSM hillshade from 2017. Note there were no data available in 2012. B. Timeseries of mean relative elevation change from 2009-2017 for the three subglacial lakes identified in Figure 1 (lake numbers are the same). Relative elevation change is calculated by taking the mean subglacial lake elevation anomaly from the mean elevation of a 500 m buffer surrounding the lake. Error bars represent the internal accuracy of the ArcticDEM (±0.2 m). Observation of the Landsat archive indicates that surface meltwater does not pond in the collapse basins following lake drainage, likely because of the heavily crevassed ice surface. Calculated mean relative elevation change is therefore a measure of ice-surface elevation change alone. Note, that the 2011 elevation anomaly is poorly constrained (dashed black line) and could therefore have been larger than suggested by the line.

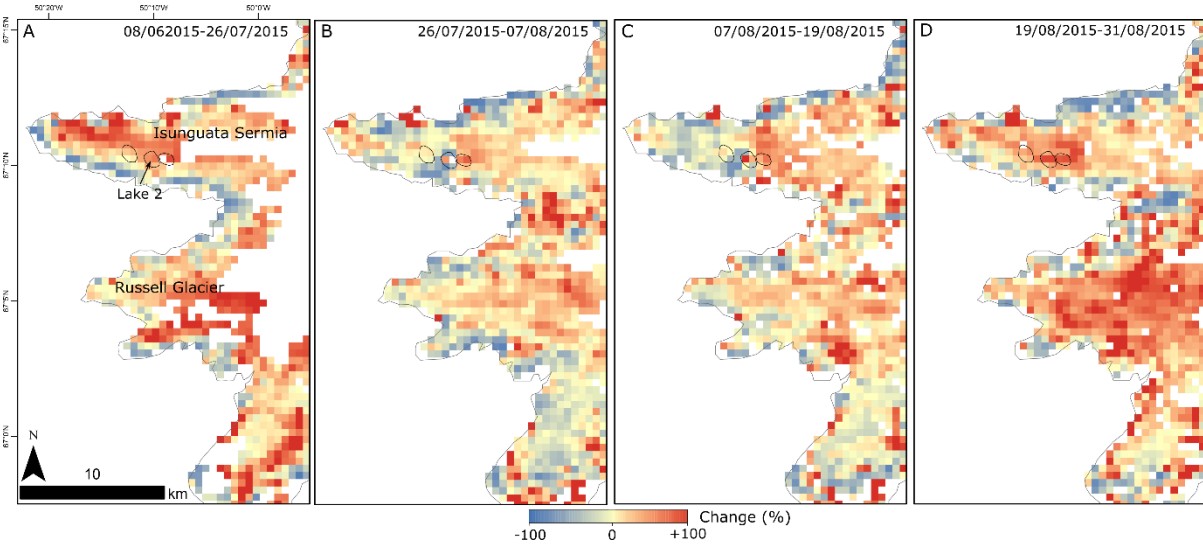

**Figure 2:** Ice dynamic response to subglacial lake drainage. Ice velocity anomalies (% change) are relative to Winter 2015 (3rd January – 5th April). Note in B and C the abrupt shift from positive to negative anomalies at the location of and downstream of Subglacial Lake 2, with the negative anomalies indicating a return to roughly winter average values during the period that it drained (between 26th July and 19th August). This coincided with a regional slowdown in ice flow. This was followed by recovery to pre-drainage ice flow velocities by 19th to 31st August (D). The strong positive anomaly over the lake in C is caused by ice flow convergence into the depression, leading to localised 'up-glacier' flow against the regional easterly flow direction.

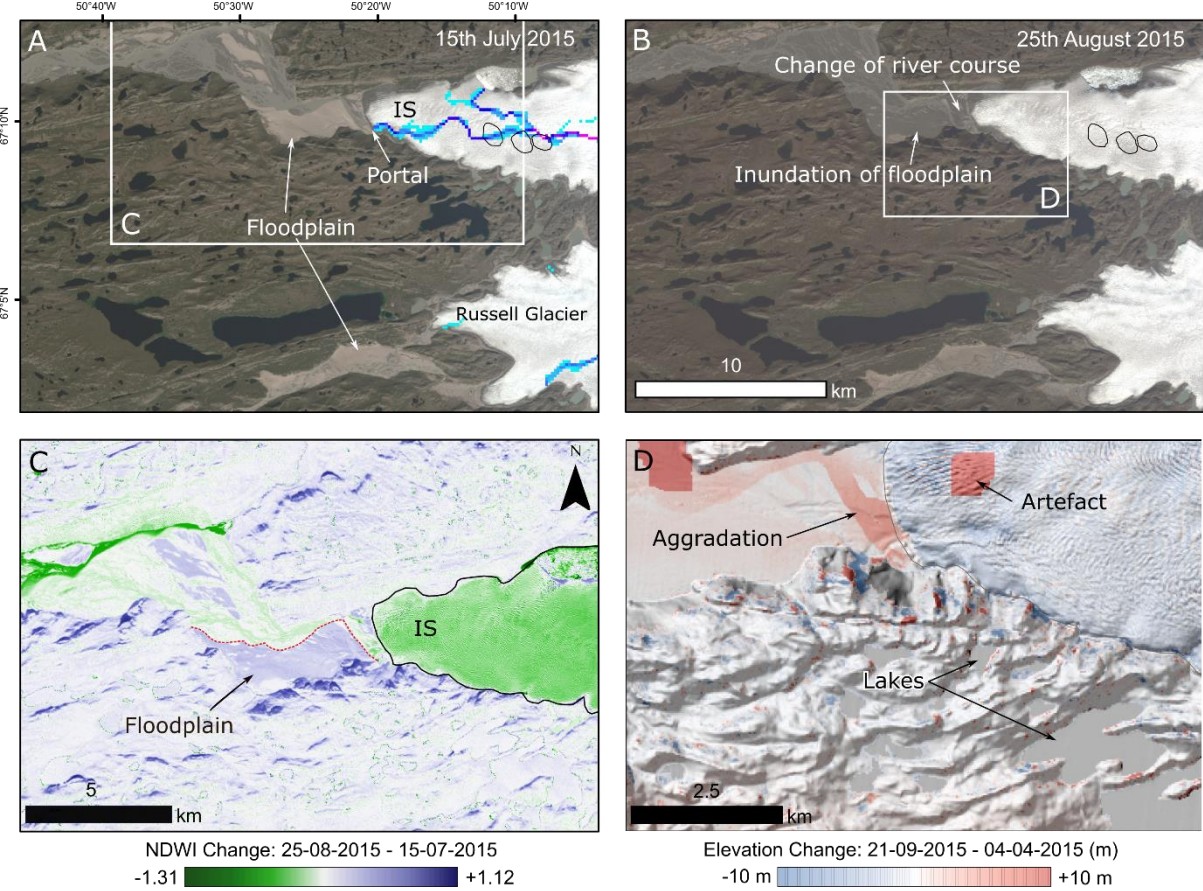

**Figure 3:** Proglacial signature of the 2015 subglacial lake drainage event. A and B are True Colour Landsat 8 images of Isunguata Sermia and the foreland before (15th July 2015) and after (25th August 2015) the drainage event. The blue/purple coloured line in A represents the predicted subglacial drainage routeway. IS = Isunguata Sermia. Note how the proglacial river changes its course and the whole floodplain becomes inundated resulting in a change of colour. This is not replicated at Leverett-Russell Glacier, ruling out a common external forcing (e.g. heavy rainfall). C is change in Normalised Difference Water Index (NDWI) between 7th July and 25th August 2015. Note the wetting (positive values) of the previous dry regions (bars and floodplain). D reveals the change in elevation from two ArcticDEM DSM tiles (co-registered to remove systematic vertical offsets using the mean vertical difference between common bedrock surfaces). There is up to 8 m of aggradation close to the glacier portal which declines with distance from the outlet. The dotted line demarcates the mapped wetted area in panel A.