# Peer review of "Brief Communication: Subglacial lake drainage beneath Isunguata Sermia, West Greenland: geomorphic and icedynamic effects"

_The Cryosphere, 2019_

## Referee Comment (RC1) · Katrin Lindbäck (Referee) · 13 Aug 2019

**GENERAL COMMENTS**

This paper presents three new observations of subglacial lakes identified from satellite surface elevation data near the margin of a land-terminating section of the western Greenland Ice Sheet. The lakes are small in size, but their location near the ice margin makes them easy study objects for future investigations, compared to subglacial lakes in the interior of the ice sheet. Subglacial lakes have only recently been identified in Greenland, compared to in Antarctica, and therefore there is a potential to study these features in more detail to understand how they interact spatially and temporarily with

the subglacial and proglacial hydrological systems. The paper is well structured and the language is fluent. I recommend publication after minor revision taking into account my general and detailed comments below. I apologize for any misunderstandings and look forward to seeing a revised version of the paper. My main comments and suggestions for improvements are:

1. I find the title does not reflect the paper content in a proper way; it refers to "outbursts floods". There is only one such event documented in Fig. 3. Are similar outburst floods observed for the other two lake drainage events? How common are these kind of flooding observations in satellite data from Isunnguata Sermia? Could the observed event coincide with supraglacial lake drainage events upglacier? Also, there are no drainage data presented to verify the qualitative observations from the satellite image. I would like to see some description of these caveats in the discussion section.

2. Since subglacial lakes are relatively new findings in Greenland, it would be nice with some more review of previous studies in the introduction linked to the discussion. Are the lakes in this study a new type of subglacial lake in Greenland or have they been observed elsewhere?

3. The methods are described shortly at the end of the introduction. I believe not all readers are familiar with these data and methods. Therefore, a methods description could be added in supplements. In this description, a short review on how subglacial lakes have been found in previous studies could be included.

SPECIFIC COMMENTS

Title:

The usage of plural of "outbursts floods" needs to be reflected in the paper. Only one observation of an outburst flood is presented for Lake 2 in Fig. 3. Are there additional satellite images showing outbursts floods for lake 1 and 3? If there is not room for additional figures in the paper, they could be included in supplements. Or the title

could be changed to reflect the content of the paper.

Introduction:

L28: "Shallow hydraulic gradients" sounds confusing to me when referring to water, maybe write "low hydraulic gradients"?

L37: I suggest replacing the word "significant" since it is a statistical term.

L37-39: This sentence holds a lot of information and is a bit unclear to me, e.g. please clarify what you mean with "surface imprints". Do they not often coincide with subglacial depressions and potential subglacial lakes?

L40-44: Could you add some more review on these findings? Also, you mentioned three types of lakes here. Are the ones described in this paper a new type of lake (marginal lakes that fill over several years)? Please mention in the discussion.

L46: Do you have a reference for the statement "..is thought unlikely. . ." or is it from the references above? If so, please move the references to the end of the sentence.

L47: Add a reference for the Landsat data.

L51: Vertical accuracy?

L52: How were the DSMs corrected against filtered IceSAT data? Did you do this? This sentence is a bit unclear.

L53: Please describe in more detail how NDWI is used. Is there a reference to this method?

L50-55: These sentences describe methods and do not fit very well in the introduction. I suggest to move them to the next section and rename it to "Methods and Observations" or similar. Also, it would be clarifying with a last sentence in the introduction describing the objective and aim of the study.

Discussion:

[Figure]

L91-93: Levett and Russell Glacier have another subglacial drainage catchment than Isunnguata Sermia, so these two are not necessarily connected. Have you checked with other potential sources of subglacial water upglacier, such as supraglacial lake drainage events?

L95: Wrong reference, please correct. The subglacial hydrological analysis was made in the Lindbäck et al. (2015) GRL paper (doi:10.1002/2015GL065393).

L98: "...one drainage event _each_ over..."

L100: How were the uncertainties ($\pm$) of each lake volume change determined?

L106: "largest and best-constrained _lake_..."?

L112-114: I don't follow this statement "recharge were similar over winter and summer". February is a winter month, please rephrase the comparison periods. Also, the plot in Fig. 2 for Lake 2 looks steeper in summer 2016 than in winter, suggesting a faster refill in summer. The other two plots do not have high enough temporal resolution in summer to support the statement.

L118: What do you mean with "in close proximity"? Are you referring to other lakes than these three? Please clarify the sentence.

L120: Do you have a reference for this modelling work?

L121: As mentioned earlier. How about supraglacial drainage events upglacier? Can these be ruled out?

L133: One difficulty with future studies of these lakes, is that it is hard to predict when the lakes will drain in the data (almost no observed filling/elevation change before the drainage events in Fig, 2). Any recommendations regarding this?

Conclusions:

L154: As mentioned earlier, I would avoid using the term "significant" for qualitative

data.

Figure 1:

North arrow and spatial reference are missing. I find it difficult to see the color differences, eg. 1 m compared to 10 m change. Also, is it possible to provide exact date for the images used in the subtraction? Makes it easier to reproduce the results.

Figure 2:

Nice figure.

Figure 3:

North arrow and spatial reference are missing. Fig. A: Define IS in the caption. Fig C: Why is the ice green? Fig. D: Why are the lakes blue? Are they masked out or have they lowered 10 m in elevation? Seems unlikely.

---

## Referee Comment (RC2) · Martin Lüthi (Referee) · 26 Aug 2019

General comments

The manuscript is well written and reports on some very interesting observations. The main shortcoming is a complete lack of the investigation and discussion of ice-dynamical effects. While a numerical study is clearly outside the scope of this paper, the DEM and satellite velocity data products could be easily investigated to answer some important questions.

A sublgacial lake of a lateral extent of twice the ice thickness will strongly affect the

surface topography and the ice flow field. Is there any evidence of a flat surface over the lake (this should be readily visible form the DEM)? Is the surface structure changing after the drainage events, e.g. a downstream buldge, or crevasse zones?

The ice flow field would also be greatly affected by uncoupling from the bed of such a big area. Is there indication of increased lateral crevassing, or a compressional zone including a surface bulge at the downstream end of the lake? Are ice velocities higher over the area of the subglacial lake? Are velocities changing during drainage and refilling?

Minor comments

26 "channel melt rates" (it is important to distinguish this from surface melt).

33 "surface melt water"

41 This is somewhat problematic, as the lakes are at the ice bottom, which is not above the ELA. Their locations are at positions in the accumulation area, where the *surface* is above the ELA, or simply, above the EL.

62 Were these anomalies stable in space, or moving with the ice?

72 An indication of the ice thickness above these features is needed.

115 The change in surface elevation is only discussed in terms of subglacial water storage. Another cause could be ice compression by horizontally convergent ice flow. Can this be ruled out by the surface velocity field?

Figure 1: Years are barely readable. Better underlay the numbers by white background. Also describe in the caption that the "ice dammed lake (white background)" is visible on the surface.

Figure 2: The black line should be broken at 2011, as the anomaly might have been much lower than suggested by the line.

[Figure]

---

## Author Comment (AC2) · 19 Sep 2019

We thank Reviewer 2 for their comments on our paper. Their suggestion to expand on the ice-dynamic impact had led to us carrying out additional ice velocity analysis for the 2015 drainage event using feature tracking of radar (Sentinel 1a) satellite data. The results suggest a slowdown in ice flow during the drainage event, and we have now included this in the paper through the addition of a figure, and methods, results and discussion. Our responses can be found below. Reviewer comments are in black and replies in blue.

On behalf of all co-authors,

Kind Regards,

Stephen Livingstone

**Reviewer 2**

General Comments

The manuscript is well written and reports on some very interesting observations. The main shortcoming is a complete lack of the investigation and discussion of ice dynamical effects. While a numerical study is clearly outside the scope of this paper, the DEM and satellite velocity data products could be easily investigated to answer some important questions.

A subglacial lake of a lateral extent of twice the ice thickness will strongly affect the surface topography and the ice flow field. Is there any evidence of a flat surface over the lake (this should be readily visible form the DEM)? Is the surface structure changing after the drainage events, e.g. a downstream bulge, or crevasse zones?

In terms of a surface expression of lake drainages, unlike in other examples of Greenland subglacial lake drainages, we do not see any evidence of a compressional zone or increased lateral crevassing at the downstream end of the lake (see Figure 1 below which shows hillshaded surface topography before and after the 2015 drainage event). The only evidence seems to be lowering of the surface above the lake, which you can see by the shadow at the downstream (left-hand) end of the lake. The ice surface above the lakes is also not flat (Figure 1). We suggest a number of reasons why this could be the case: (1) the relatively modest thickness of ice (approx. 400 m beneath lake 2) and small predicted size of lakes (<1 km$^2$), which are likely maximum estimates given the influence of bridging stresses and the viscosity of the ice on the transmission of the effects of lake drainage to the surface; (2) unlike subglacial lakes observed elsewhere (e.g. Antarctica and further from the margin beneath the Greenland Ice Sheet) these are in a relatively confined outlet lobe and close to an ice-dammed lake. Due to the relatively deep depressions and lakes at the lateral ice margins, the ice flow in part 'peels' off to either side of the main glacier trunk. This creates a complex pattern of crevassing that may hide the (relatively subtle for the reasons outlined above) influence of the lake; (3) if the bed was relatively rough, coupled with the small sizes of the lakes, the bed would have a large effect relative to the lake, whereas in Antarctica where lakes are often an order of magnitude larger, the lake 'slippery spot' may dominate; and (4) complex surface hydrology, emergence of debris-rich layers at the surface and localised meteorological factors (katabatic winds moving emergent dust around, and strong solar insolation with resulting albedo variations) results in a complex ice surface topography with almost ubiquitous 4-5 m relief that is independent of crevasse formation.

[Figure]

*Figure 1: ArcticDEM hillshaded DSMs before (2nd August) and after (21st September) the 2015 subglacial lake drainage event. The lake that drained during this period is the middle of the polygons. The drainage is picked out by a drop in ice-surface elevation. Note how the surface is not completely flat.*

The ice flow field would also be greatly affected by uncoupling from the bed of such a big area. Is there indication of increased lateral crevassing, or a compressional zone including a surface bulge at the downstream end of the lake? Are ice velocities higher over the area of the subglacial lake? Are velocities changing during drainage and refilling?

Good point. Although we could not identify an increase in lateral crevassing, compression (e.g. a surface bulge) or locally higher velocities over the subglacial lakes, we did identify an ice-dynamic response during the 2015 drainage event, although this is complicated by a regional slowdown that occurred during the same time. We used Sentinel 1a radar data (12-day repeat image pairs) to calculate ice velocity from feature and speckle tracking (adopting the same method published in Tuckett et al., (2019)). Anomalies were calculated relative to the 2015 winter mean, and revealed a distinctive and abrupt slowdown to winter values immediately downstream of the lake (relative to upstream, where values were positive) over the period during which it drained. We believe this is the first evidence for a net slowdown in ice flow following a subglacial lake drainage and have therefore added in a new methods section as an Appendix to detail the ice velocity methods; combined figures 1 and 2 (which focus on ice-surface elevation change) and added a new figure 2 where we show the ice velocity anomalies; and expanded both the Observation and Discussion sections to include a description of the ice velocity response and then some discussion on why a net slowdown is possible in the context of these lakes. We think this has added substantially to the paper, so thank the reviewer for his suggestion.

*Tuckett, P.A., Ely, J.C., Sole, A.J., Livingstone, S.J., Davison, B.J., Melchior van Wessem, J., Howard, J. Large and rapid accelerations of Antarctic Peninsula outlet glaciers driven by surface melt. Nature Communications, 2019.*

Minor comments
26 "channel melt rates" (it is important to distinguish this from surface melt).

Done

33 "surface melt water"

Not all water in subglacial lakes will be from surface melting (e.g. basal frictional and geothermal melting) and we therefore prefer to leave this as just meltwater.

41 This is somewhat problematic, as the lakes are at the ice bottom, which is not above the ELA. Their locations are at positions in the accumulation area, where the *surface* is above the ELA, or simply, above the EL.

Good point. We have rephrased as per the reviewers last suggestion, EL.

62 Were these anomalies stable in space, or moving with the ice?

Yes, good point, these anomalies are stable in space, i.e. they do not migrate down ice through time. We have added this point to the methods and observation section – *"Timeseries of relative elevation change for each anomaly, which are not advected towards the margin, were calculated from sub-annual ArcticDEM DSMs by subtracting the mean ice-surface elevation of the anomaly from the mean elevation of a 500 m buffer around it (Fig. 1b)."*

72 An indication of the ice thickness above these features is needed.

Rough ice thicknesses for each anomaly have been added to the descriptions.

115 The change in surface elevation is only discussed in terms of subglacial water storage. Another cause could be ice compression by horizontally convergent ice flow. Can this be ruled out by the surface velocity field?

If we understand you correctly, we do not think this likely for a number of reasons. (1) the anomalies are all circular to ovoid in form, which is consistent with a ponded water body rather than horizontally convergent ice flow, which we might expect to produce a more flow-parallel, linear feature; (2) horizontally convergent ice flow would produce the rise in ice surface but does not account for the drop in elevation; (3) the drop in elevation in the 2015 example coincides with an outburst flood and proglacial sediment accumulation, which we suggest must have been caused by a rapid drainage event; and (4) the anomalies are stable in space (i.e. they do not migrate down ice).

Figure 1: Years are barely readable. Better underlay the numbers by white background. Also describe in the caption that the "ice dammed lake (white background)" is visible on the surface.

We have extended the caption to mention the ice-dammed lake. We have added a white background to all the numbers and text.

Figure 2: The black line should be broken at 2011, as the anomaly might have been much lower than suggested by the line.

We have made the line dashed at 2011 to indicate that the anomaly could have been much greater, and also added a comment in the caption. Note this is now Figure 1b to account for the new ice dynamic work and figure.

---

## Author Response (AR1)

**Dear Editor**

We thank both reviewers for their comments on our paper. Their suggestions are both on point and have helped to improve the paper substantially. In particular, we made changes in response to all major comments, including modifying the title to better reflect the outcomes of the study; analysed velocity data using feature tracking of Sentinel 1a radar imagery and included sections on the ice dynamic response to the 2015 subglacial lake drainage event (we believe we show for the first time

that subglacial lake drainage can cause a net slowdown, which we believe is a response to the glaciological context of the lakes); extended the methods, which have now been moved to a supplementary section (Appendix A); and included more background on subglacial lakes in Graenland which we then raturn to in the discussion

10 Greenland, which we then return to in the discussion.

Our responses can be found below. Reviewer comments are in black and replies in blue.

On behalf of all co-authors,

Kind Regards,

Stephen Livingstone

**15**

20

25

5

**Reviewer 1**

This paper presents three new observations of subglacial lakes identified from satellite surface elevation data near the margin of a land-terminating section of the western Greenland Ice Sheet. The lakes are small in size, but their location near the ice margin makes them easy study objects for future investigations, compared to subglacial lakes in the interior of the ice sheet. Subglacial lakes have only recently been identified in Greenland, compared to in Antarctica, and therefore there is a potential to study these features in more detail to understand how they interact spatially and temporarily with the subglacial and proglacial hydrological systems. The paper is well structured and the language is fluent. I recommend publication after minor revision taking into account my general and detailed comments below. I apologize for any misunderstandings and look forward to seeing a revised version

of the paper. My main comments and suggestions for improvements are:

 I find the title does not reflect the paper content in a proper way; it refers to "outbursts floods". There is only one such event documented in Fig. 3. Are similar outburst floods observed for the other
 two lake drainage events? How common are these kind of flooding observations in satellite data from Isunnguata Sermia? Could the observed event coincide with supraglacial lake drainage events upglacier? Also, there are no drainage data presented to verify the qualitative observations from the satellite image. I would like to see some description of these caveats in the discussion section.

We agree this is misleading and have modified the title to "Brief Communication: Subglacial lake
drainage beneath Isunguata Sermia, West Greenland: geomorphic and ice-dynamic effects", which we think better reflects the key findings of the paper, and takes into account the new data included in the revised paper. We tried to identify outburst floods associated with the other two subglacial lake drainage events, but could not find any conclusive evidence. The satellite and DEM archive is patchier for these two earlier events (particularly in-front of Isunguata Sermia), making it more difficult to discern any outburst events (either from NDWI or elevation change). The 2015 event also seems to have been the largest of the three, which may be one reason why we were able to clearly identify its downstream response.

2. Since subglacial lakes are relatively new findings in Greenland, it would be nice with some more review of previous studies in the introduction linked to the discussion. Are the lakes in this study a new type of subglacial lake in Greenland or have they been observed elsewhere?

Good points, thanks. We have expanded the introduction section to include more detail on how

subglacial lakes in Greenland have been identified to date (see also comment below). We have also expanded the discussion section, adding in a new paragraph where we link back to the introduction, including the three main subglacial lake types from the Bowling et al. (2019) paper and the potential for water storage to delay transfer to the margin and influence downstream ice dynamics.

3. The methods are described shortly at the end of the introduction. I believe not all readers are familiar with these data and methods. Therefore, a methods description could be added in supplements. In this description, a short review on how subglacial lakes have been found in previous studies could be included.

55 We have expanded the methods, which we have now moved to an Appendix as also suggested, to include more details of the NDWI method (as also recommended in the specific comments). We have also added information on how subglacial lakes in previous studies were identified in the introduction where we review previous subglacial lake research.

**SPECIFIC COMMENTS**

60 Title:

45

50

The usage of plural of "outbursts floods" needs to be reflected in the paper. Only one observation of an outburst flood is presented for Lake 2 in Fig. 3. Are there additional satellite images showing outbursts floods for lake 1 and 3? If there is not room for additional figures in the paper, they could be included in supplements. Or the title could be changed to reflect the content of the paper.

65 Done. See reply to major comment above re. title.

Introduction:

L28: "Shallow hydraulic gradients" sounds confusing to me when referring to water,

maybe write "low hydraulic gradients"?

Done.

80

70 L37: I suggest replacing the word "significant" since it is a statistical term.

Changed to "important".

L37-39: This sentence holds a lot of information and is a bit unclear to me, e.g. please clarify what you mean with "surface imprints". Do they not often coincide with subglacial depressions and potential subglacial lakes?

75 By 'surface inputs', we refer to the input of surface meltwater to the bed, which is a key component of the subglacial drainage system in Greenland, relative to Antarctica. We have clarified this to "surface meltwater inputs".

L40-44: Could you add some more review on these findings? Also, you mentioned three types of lakes here. Are the ones described in this paper a new type of lake (marginal lakes that fill over several years)? Please mention in the discussion.

Please see reply to major comment above.

L46: Do you have a reference for the statement "...is thought unlikely: : :" or is it from the references above? If so, please move the references to the end of the sentence.

We have added a reference (Bowling et al. 2019) to support this finding.

85 L47: Add a reference for the Landsat data.

We have added the following in brackets "(distributed by the U.S. Geological Survey - https://earthexplorer.usgs.gov/)"

L51: Vertical accuracy?

This is the horizontal accuracy of the ArcticDEM DSMs, and we have corrected to make this clear.

90 L52: How were the DSMs corrected against filtered IceSAT data? Did you do this? This sentence is a bit unclear.

The mean offset between ArcticDEM swaths and coincident IceSAT elevations is provided in the ArcticDEM metadata, and so this correction when available could just be applied directly to the ArcticDEM tile. This is specified in our new Appendix A. Datasets and Methods.

95 L53: Please describe in more detail how NDWI is used. Is there a reference to this method?

We have expanded the section on NDWI to explain its use and pre-processing steps, and also now include a reference - Zhao et al. (2018) - although to stay within the limit of a brief communication we have had to delete a reference elsewhere to accommodate this.

Zhao, H., Chen, F., and Zhang, M. A systematic extraction approach for mapping glacial lakes in
high mountain regions of Asia. IEEE Journal of Selected Topics in Applied Earth Observations and Remote Sensing, 11(8), 2788-2799. doi: 10.1109/JSTARS.2018.2846551, 2018.

L50-55: These sentences describe methods and do not fit very well in the introduction. I suggest to move them to the next section and rename it to "Methods and Observations" or similar. Also, it would be clarifying with a last sentence in the introduction describing the objective and aim of the study.

**105** To also account for the ice velocity methods and additional information on NDWI and uncertainties we have we have moved the last few sentences to a new, expanded - Appendix A. Datasets and Methods.

Discussion:

L91-93: Leverett and Russell Glacier have another subglacial drainage catchment than Isunnguata
 Sermia, so these two are not necessarily connected. Have you checked with other potential sources of subglacial water upglacier, such as supraglacial lake drainage events?

Certainly, during the period of late August and early September, when these subglacial lakes drained, there will also have been a number of supraglacial lake drainage events, and this is evident from checking the available satellite imagery. However, we do not think a supraglacial lake drainage event

115 a likely cause of our proglacial observations given that the outburst flood coincides with the timing of ice-surface elevation anomaly 2. In addition, supraglacial lake drainages are relatively common along this western margin of Greenland, but major outburst floods characterised by rapid aggradation of up to 8 m of sediment are not.

L95: Wrong reference, please correct. The subglacial hydrological analysis was made in the Lindbäck
et al. (2015) GRL paper (doi:10.1002/2015GL065393).

Good point. However, we are at the limit of the number of references allowed and so we have chosen to delete the wrong reference and just use Chu et al. (2016) as an example, given this is also used elsewhere in the manuscript.

L98: ": : : : one drainage event \_each\_ over: : : "

125 Done

L100: How were the uncertainties  $(\pm)$  of each lake volume change determined?

The uncertainties of each lake volume change were determined by multiplying the internal error of the ArcticDEM by the surface area both before and after drainage and adding the errors together in quadrature. We have added a sentence to this effect in the Appendix.

130 L106: "largest and best-constrained \_lake\_:::"?

Done

135

L112-114: I don't follow this statement "recharge were similar over winter and summer". February is a winter month, please rephrase the comparison periods. Also, the plot in Fig. 2 for Lake 2 looks steeper in summer 2016 than in winter, suggesting a faster refill in summer. The other two plots do not have high enough temporal resolution in summer to support the statement.

We agree that we have little data supporting this statement and discussion and so have deleted this section on lake recharge between summer and winter.

L118: What do you mean with "in close proximity"? Are you referring to other lakes than these three? Please clarify the sentence.

140 We have clarified our meaning here by adding "...the three subglacial lakes are..."

L120: Do you have a reference for this modelling work?

This is poorly phrased – we are actually referring to the hydrological routing analysis (Shreve equation, assuming ice overburden = water pressure) here, which is shown in Fig. 3a, and have rephrased accordingly.

145 L121: As mentioned earlier. How about supraglacial drainage events upglacier? Can these be ruled out?

See comment above, we believe a supraglacial lake drainage event causing our proglacial observations is unlikely.

L133: One difficulty with future studies of these lakes, is that it is hard to predict when the lakes willdrain in the data (almost no observed filling/elevation change before the drainage events in Fig, 2).Any recommendations regarding this?

This is currently a challenge as we only have one drainage event per lake and so we cannot calculate the recurrence interval. In addition, the lakes seem to fill and then remain stable for some time before then draining again and so we cannot estimate based on how full the lake is (i.e. they do not seem to reach a drainage threshold). Hopefully, as the 2018 and then 2019 ArcticDEM timestamped data are

released we will capture repeat drainage events that will help us to begin to answer that question.

Conclusions:

155

L154: As mentioned earlier, I would avoid using the term "significant" for qualitative data.

We have deleted the word "significant" here.

160 Figure 1:

North arrow and spatial reference are missing. I find it difficult to see the color differences, eg. 1 m compared to 10 m change. Also, is it possible to provide exact date for the images used in the subtraction? Makes it easier to reproduce the results.

We have added both a north arrow and spatial reference. The images do not have exact dates, as they are actually down-sampled (to 50 m) composites produced by merging (using median values where there is overlap) all the DSMs available in that particular year. This was done to produce a more consistent DSM of larger spatial extent to better identify large-scale changes, and was needed as the timestamped DSMs are rather patchy. We have now added some brief details to the caption detailing how the DSMs were produced and making this clear.

170 Figure 2:

Nice figure.

Thanks

Figure 3:

North arrow and spatial reference are missing. Fig. A: Define IS in the caption. Fig C: Why is the icegreen? Fig. D: Why are the lakes blue? Are they masked out or have they lowered 10 m in elevation?Seems unlikely.

We have added a north arrow and graticule. IS is defined in the caption in the fourth sentence. There are two possible reasons why the ice is green (-ve) in the NDWI plot of Figure 3C. The ice might be drier in the second image, thus reducing the NDWI value and therefore on the change in NDWI

180 figure, indicate a reduction in water content; and/or a change in sun angle can influence the brightness of the ice and therefore have a slight impact on the NDWI values. The lakes are blue because in one of the DSMs there are NoData values (-9999). This has now been rectified, with these values turned to Null.

**185 **Reviewer 2**

190

**General Comments**

The manuscript is well written and reports on some very interesting observations. The main shortcoming is a complete lack of the investigation and discussion of ice dynamical effects. While a numerical study is clearly outside the scope of this paper, the DEM and satellite velocity data products could be easily investigated to answer some important questions.

A subglacial lake of a lateral extent of twice the ice thickness will strongly affect the surface topography and the ice flow field. Is there any evidence of a flat surface over the lake (this should be readily visible form the DEM)? Is the surface structure changing after the drainage events, e.g. a downstream bulge, or crevasse zones?

- 195 In terms of a surface expression of lake drainages, unlike in other examples of Greenland subglacial lake drainages, we do not see any evidence of a compressional zone or increased lateral crevassing at the downstream end of the lake (see Figure 1 below which shows hillshaded surface topography before and after the 2015 drainage event). The only evidence seems to be lowering of the surface above the lake, which you can see by the shadow at the downstream (left-hand) end of the lake. The
- 200 ice surface above the lakes is also not flat (Figure 1). We suggest a number of reasons why this could be the case: (1) the relatively modest thickness of ice (approx. 400 m beneath lake 2) and small

predicted size of lakes (

*Figure 1: ArcticDEM hillshaded DSMs before (2nd August) and after (21st September) the 2015 subglacial lake drainage event. The lake that drained during this period is the middle of the polygons. The drainage is picked out by a drop in ice-surface elevation. Note how the surface is not completely flat.*

220

The ice flow field would also be greatly affected by uncoupling from the bed of such a big area. Is there indication of increased lateral crevassing, or a compressional zone including a surface bulge at the downstream end of the lake? Are ice velocities higher over the area of the subglacial lake? Are velocities changing during drainage and refilling?

- 225 Good point. Although we could not identify an increase in lateral crevassing, compression (e.g. a surface bulge) or locally higher velocities over the subglacial lakes, we did identify an ice-dynamic response during the 2015 drainage event, although this is complicated by a regional slowdown that occurred during the same time. We used Sentinel 1a radar data (12-day repeat image pairs) to calculate ice velocity from feature and speckle tracking (adopting the same method published in
- 230 Tuckett et al., (2019)). Anomalies were calculated relative to the 2015 winter mean, and revealed a distinctive and abrupt slowdown to winter values immediately downstream of the lake (relative to upstream, where values were positive) over the period during which it drained. We believe this is the first evidence for a net slowdown in ice flow following a subglacial lake drainage and have therefore

added in a new methods section as an Appendix to detail the ice velocity methods; combined figures 1

- and 2 (which focus on ice-surface elevation change) and added a new figure 2 where we show the ice velocity anomalies; and expanded both the Observation and Discussion sections to include a description of the ice velocity response and then some discussion on why a net slowdown is possible in the context of these lakes. We think this has added substantially to the paper, so thank the reviewer for his suggestion.
- 240 Tuckett, P.A., Ely, J.C., Sole, A.J., Livingstone, S.J., Davison, B.J., Melchior van Wessem, J., Howard, J. Large and rapid accelerations of Antarctic Peninsula outlet glaciers driven by surface melt. Nature Communications, 2019.

Minor comments

26 "channel melt rates" (it is important to distinguish this from surface melt).

245 Done

33 "surface melt water"

Not all water in subglacial lakes will be from surface melting (e.g. basal frictional and geothermal melting) and we therefore prefer to leave this as just meltwater.

41 This is somewhat problematic, as the lakes are at the ice bottom, which is not above the ELA.
250 Their locations are at positions in the accumulation area, where the \*surface\* is above the ELA, or simply, above the EL.

Good point. We have rephrased as per the reviewers last suggestion, EL.

62 Were these anomalies stable in space, or moving with the ice?

Yes, good point, these anomalies are stable in space, i.e. they do not migrate down ice through time.
We have added this point to the methods and observation section – "Timeseries of relative elevation change for each anomaly, which are not advected towards the margin, were calculated from sub-annual ArcticDEM DSMs by subtracting the mean ice-surface elevation of the anomaly from the mean elevation of a 500 m buffer around it (Fig. 1b)."

72 An indication of the ice thickness above these features is needed.

260 Rough ice thicknesses for each anomaly have been added to the descriptions.

115 The change in surface elevation is only discussed in terms of subglacial water storage. Another cause could be ice compression by horizontally convergent ice flow. Can this be ruled out by the surface velocity field?

If we understand you correctly, we do not think this likely for a number of reasons. (1) the anomalies are all circular to ovoid in form, which is consistent with a ponded water body rather than horizontally convergent ice flow, which we might expect to produce a more flow-parallel, linear feature; (2) horizontally convergent ice flow would produce the rise in ice surface but does not account for the drop in elevation; (3) the drop in elevation in the 2015 example coincides with an outburst flood and proglacial sediment accumulation, which we suggest must have been caused by a rapid drainage event; and (4) the anomalies are stable in space (i.e. they do not migrate down ice).

Figure 1: Years are barely readable. Better underlay the numbers by white background. Also describe in the caption that the "ice dammed lake (white background)" is visible on the surface.

We have extended the caption to mention the ice-dammed lake. We have added a white background to all the numbers and text.

Figure 2: The black line should be broken at 2011, as the anomaly might have been much lower than suggested by the line.

We have made the line dashed at 2011 to indicate that the anomaly could have been much greater, and also added a comment in the caption. Note this is now Figure 1b to account for the new ice dynamic work and figure.

**300 Brief Communication: Outburst floods triggered by periodic drainage of subglacial lakes, Isunguata Sermia, West Greenland Brief Communication: Subglacial lake drainage beneath Isunguata Sermia, West Greenland: geomorphic and ice-dynamic effects**

**305 Stephen J. Livingstone1, Andrew J. Sole1, Robert D. Storrar2, Devin Harrison3, Neil Ross3, Jade Bowling4**

1Department of Geography, University of Sheffield, Sheffield, UK

2Department of the Natural and Built Environment, Sheffield Hallam University, UK

3School of Geography, Politics and Sociology, Newcastle University, Newcastle upon Tyne, UK

310 4Lancaster Environment Centre, Lancaster University, Lancaster, UK

Correspondence to: Stephen J. Livingstone (s.j.livingstone@sheffield.ac.uk)

**Abstract (100 words)**

We report three active subglacial lakes within 2—\_\_km of the lateral margin of Isunguata Sermia, West
 Greenland, identified by differencing time-stamped ArcticDEM strips. Each lake underwent one drainage-refill event between 2009 and 2017, with two lakes draining in <1 month during betweenin August 2014 and August 2015, and all three characterised by 2 3 year refill periods. The 2015 drainage caused a net~1-month down-glacier slowdown in ice-flow and flooded the foreland, aggrading 8 m of the proglacial channel by 8 m, confirming. Thise proglacial flooding confirms the ice-surface elevation anomalies as subglacial water bodies and demonstrating demonstrates how subglacial lake their drainages can significantly modify proglacial environments. These subglacial lakes offer accessible targets for future geophysical investigations and exploration.</li>

**1. Introduction**

- 325 Meltwater beneath the Greenland Ice Sheet is sourced from geothermal and frictional melting, and via the input of surface meltwater through englacial pathways. This meltwater drains towards the ice sheet margin through a complex network of inefficient and efficient drainage routes (Davison et al., 2019). Spatial and temporal variations in drainage structure are controlled by the hydraulic gradient and meltwater flux. Steeper hydraulic gradients and higher meltwater fluxes close to the ice margin lead to greater ice-channel melt rates and promote the formation of efficient channels, which can extend up to 40 km inland and evolve on seasonal timescales in response to surface meltwater inputs (Chandler et al., 2013). Shallow-Low hydraulic gradients and lower-smaller meltwater fluxes dominated by subglacial meltwater sources tend to be associated with more inefficient drainage configurations further inland (Doyle et al., 2014).
- Storage of water in firn (Forster et al., 2013), damaged englacial ice (Kendrick et al., 2018) and both supraglacial (Selmes et al., 2011) and subglacial lakes (Palmer et al., 2013; Oswald et al., 2018; Bowling et al., 2019) can delay the drainage of meltwater through the ice sheet to the ocean, while the rapid drainage of stored water can overwhelm the drainage system and perturb ice flow (e.g. Das et al., 2008). Storage and drainage of supraglacial lakes have been well-documented (e.g. Selmes et al., 2011), but the volume of water stored subglacially, and the lakes' residence times of the lakes, and the wider influence on the subglacial hydrological system and ice flow is poorly understood. Although subglacial lakes are expected to be a less significantimportant component of the hydrological system compared with Antarctica (e.g. Siegfried & Fricker, 2018) due to steeper hydraulic gradients, dominance of surface meltwater inputs and more efficient subglacial

water routing, 1000s of subglacial lakes have been predicted and over 50 identified beneath the Greenland Ice Sheet (Livingstone et al., 2013; Bowling et al., 2019). This includes stable lakes above the Equilibrium Line Altitude (ELA) but away from the interior identified from airborne radio-echo sounding (Palmer et al., 2013;

- 345 Altitude (ELA) but away from the interior identified from airborne radio-echo sounding (Palmer et al., 2013; Oswald et al. 2018; Bowling et al., 2019); hydrologically active lakes recharged by surface meltwater near the EL, determined from surface elevation change measurements derived from repeat high-resolution Digital Surface Models (DSMs) and ICESat elevation data (Palmer et al., 2015; Willis et al., 2015; Bowling et al., 2019)near the ELA recharged by surface meltwater; and small seasonally active lakes subglacial water bodies
- 350 below the ELA\_-which form during winter and drain during the melt season identified from repeat airborne radio-echo sounding (Palmer et al., 2013, 2015; Howat et al., 2013; Willis et al., 2015; Chu et al., 2016; Oswald et al., 2018; Bowling et al., 2019).

Whilst seasonal water storage is thought to be common below the ELA (e.g. Chu et al., 2016; Kendrick et al., 2018), longer term subglacial lake storage is thought unlikely due to the development of efficient channels and the associated increase in hydrological connectivity each melt season (Bowling et al., 2019). In this paper we acquired multi-temporal ArcticDEM Digital Surface Models (DSMs) (Noh & Howat, 2015) and Landsat 7 and 8

- satellite imagery between 2009 and 2017 (distributed by the U.S. Geological Survey https://earthexplorer.usgs.gov/) to identify three active subglacial lakes on a reverse bed-slope beneath Isunguata Sermia, West Greenland (67°10' N, 50°12'W) (Fig. 1). The ArcticDEM DSMs were

---

## Author Response (AR2)

**Dear Editor,**

We are delighted that you have accepted our paper subject to corrections. Below you can see the response and track-changed corrections. Comments are in black and response in blue.

On behalf of all co-authors,

Kind Regards,

Stephen Livingstone

Your responses to the comments and revisions are satisfactory, and I am happy to accept this paper with a few technical corrections. Line numbers below cite those in the editorial-tracking version.

- Line 386 and Line 402: change "S1 methods" to "Appendix"

Done.

- Line 425: change "for anomalies" to "for subglacial lakes"

Done

- Figure captions include some statements duplicated in the main text. In most cases, these statements can be removed from the captions.

We have been through and removed all sentences that duplicate text in the main manuscript.

[revised manuscript text omitted]